# Interface-acting nucleotide controls polymerization dynamics at microtubule plus- and minus-ends

Lauren A McCormick[1†], Joseph M Cleary[2†], William O Hancock[2]*, Luke M Rice[1]*

[1]Department of Biophysics and Biochemistry, the University of Texas Southwestern Medical Center, Dallas, United States; [2]Department of Biomedical Engineering, Pennsylvania State University, State College, United States

*For correspondence:
Luke.Rice@UTSouthwestern.edu
(LMR);
woh1@psu.edu (WOH)

[†]These authors contributed
equally to this work

Competing interest: The authors
declare that no competing
interests exist.

Reviewing Editor: Kassandra
M Ori-McKenney, University of
California, United States

**Abstract** GTP-tubulin is preferentially incorporated at growing microtubule ends, but the biochemical mechanism by which the bound nucleotide regulates the strength of tubulin:tubulin interactions is debated. The 'self-acting' (cis) model posits that the nucleotide (GTP or GDP) bound to a particular tubulin dictates how strongly that tubulin interacts, whereas the 'interface-acting' (trans) model posits that the nucleotide at the interface of two tubulin dimers is the determinant. We identified a testable difference between these mechanisms using mixed nucleotide simulations of microtubule elongation: with a self-acting nucleotide, plus- and minus-end growth rates decreased in the same proportion to the amount of GDP-tubulin, whereas with interface-acting nucleotide, plus-end growth rates decreased disproportionately. We then experimentally measured plus- and minus-end elongation rates in mixed nucleotides and observed a disproportionate effect of GDP-tubulin on plus-end growth rates. Simulations of microtubule growth were consistent with GDP-tubulin binding at and 'poisoning' plus-ends but not at minus-ends. Quantitative agreement between simulations and experiments required nucleotide exchange at terminal plus-end subunits to mitigate the poisoning effect of GDP-tubulin there. Our results indicate that the interfacial nucleotide determines tubulin:tubulin interaction strength, thereby settling a longstanding debate over the effect of nucleotide state on microtubule dynamics.

## eLife assessment

This **important** study combines in vitro experiments with simulations to identify the mechanisms governing modulation of microtubule dynamics by GTP hydrolysis. The authors introduce a **convincing** new approach by using a mixed GDP/GMPCPP lattice and varying GDP concentration to reveal that the nucleotide at the interface of two tubulin dimers determines the strength of the interaction between two dimers. Overall, the findings will be of interest to biophysicists and cell biologists, especially in the field of microtubule biology.

## Introduction

Microtubules are dynamic polymers of αβ-tubulin that support motor-based transport of cargo through the cytoplasm and orchestrate the movement of chromosomes in dividing cells (*Akhmanova and Kapitein, 2022*; *Barlan and Gelfand, 2017*; *Cleary and Hancock, 2021*; *Gudimchuk and McIntosh, 2021*; *Prosser and Pelletier, 2017*). Microtubules grow by the addition of GTP-bound tubulin to the polymer ends. Once incorporated into the microtubule lattice, tubulins hydrolyze their bound GTP. The change in nucleotide state triggers conformational changes that weaken interactions between neighboring tubulins and ultimately results in catastrophe, the switch from growth to shrinkage

(*Bowne-Anderson et al., 2013*; *Gudimchuk and McIntosh, 2021*; *LaFrance et al., 2022*; *Manka and Moores, 2018*; *Roostalu et al., 2020*; *Seetapun et al., 2012*; *Zanic et al., 2013*). Defining the connection between nucleotide state and tubulin:microtubule binding kinetics is crucial for understanding how microtubules grow and how they transition to catastrophe. However, the mechanism by which nucleotide controls the strength of tubulin:tubulin interactions remains debated.

An early model explained the nucleotide-dependence of microtubule stability by positing that nucleotide state determines the conformation of tubulin: GTP-tubulin would form strong lattice contacts because GTP favors a 'straight' conformation compatible with the microtubule lattice, and GDP-tubulin would form weak lattice contacts because GDP favors a 'curved' conformation incompatible with the microtubule lattice (*Drechsel and Kirschner, 1994*; *Howard and Timasheff, 1986*; *Melki et al., 1989*; *Nicholson et al., 1999*; *Shearwin et al., 1994*; *Tran et al., 1997*; *Wang and Nogales, 2005*). By assuming that nucleotide controls the conformation of the tubulin to which it is bound, this model embodied a 'cis-acting' view of nucleotide action. However, subsequent work demonstrated that both GTP- and GDP-tubulin adopt the same curved conformation (*Nawrotek et al., 2011*; *Pecqueur et al., 2012*; *Rice et al., 2008*), which contradicted a core assumption of the cis-acting model. These structural findings led to the proposal of a 'trans-acting' mechanism in which the nucleotide bound to one tubulin controls the strength of its interactions with the next tubulin through direct contacts and/or by causing loop movements that lead to better polymerization contacts (*Ayaz et al., 2012*; *Buey et al., 2006*; *Nawrotek et al., 2011*; *Piedra et al., 2016*; *Rice et al., 2008*). The 'trans' mechanism is supported by the knowledge that the nucleotide binding site on β-tubulin forms part of the polymerization interface with the α-tubulin from the next subunit in the protofilament, and it is also consistent with the largest nucleotide-dependent conformational changes in the microtubule occurring in α-tubulin adjacent to the β-tubulin-bound nucleotide (*Alushin et al., 2014*; *Manka and Moores, 2018*; *Zhang et al., 2015*). However, the field has still not reached a consensus on the mechanism of nucleotide action (*Brouhard, 2015*; *Brouhard and Rice, 2018*; *Brun et al., 2009*; *Luo et al., 2023*; *Margolin et al., 2012*; *Schmidt and Kierfeld, 2021*; *Stewman et al., 2020*; *VanBuren et al., 2005*; *VanBuren et al., 2002*; *Zakharov et al., 2015*) and this persistent ambiguity about how nucleotide state influences tubulin:tubulin interactions limits our understanding of microtubule dynamics.

Microtubule plus- and minus-ends are structurally distinct: plus-ends present a β-tubulin polymerization interface that contains the exchangeable nucleotide, whereas minus-ends present an α-tubulin polymerization interface that does not expose a nucleotide. Debate over cis- and trans-acting mechanisms (reviewed in *Gudimchuk and McIntosh, 2021*) has persisted in part because most studies have focused solely on the plus-end, where two nucleotides – one bound to the terminal tubulin (the cis nucleotide), and one at the interface between the terminal tubulin and the next subunit in the microtubule lattice (the trans nucleotide) – could in principle be dictating the strength of lattice contacts. At the minus-end, by contrast, the exchangeable nucleotide of the terminal tubulin is already buried in the microtubule lattice. We reasoned that this fundamental difference between the plus- and minus-ends might provide a new way to test the conflicting mechanisms of nucleotide action. A few studies have compared plus- and minus-end dynamics (*Strothman et al., 2019*; *Tanaka-Takiguchi et al., 1998*; *Walker et al., 1988*), but only one of them sought to manipulate nucleotide state in a controlled manner (*Tanaka-Takiguchi et al., 1998*). For generality in considering both ends, we will hereafter refer to the trans mechanism as 'interface-acting', and the cis mechanism as 'self-acting'.

The goal of the present study was to determine whether self-acting or interface-acting mechanisms of nucleotide action govern the strength of tubulin:tubulin contacts in the microtubule. Our approach used simulations and experiments to compare how plus- and minus-end elongation are affected by GDP-tubulin. We first simulated microtubule elongation in mixed nucleotide states using models that implemented self- or interface-acting mechanisms. These simulations revealed a striking difference between the two mechanisms of nucleotide action: in the self-acting model, GDP-tubulin inhibited plus- and minus-ended growth to the same extent, but in the interface-acting model, GDP-tubulin disproportionately inhibited plus-ended growth. This observation was consistent with an earlier study that showed a selective suppression of plus-end elongation when GDP was included in the reaction mixture (*Tanaka-Takiguchi et al., 1998*). However, the reaction conditions in that earlier study did not suppress GTPase activity, and the consequent high frequency of plus-end catastrophe prevented an examination of how GDP affected microtubule growth rates. We tested our predictions experimentally using 'mixed nucleotide' assays (containing both slowly hydrolyzable GMPCPP and GDP) to

prevent catastrophe, allowing us to directly compare the relative effects of GDP-tubulin on plus- and minus-end growth rates. We found that plus-end growth was disproportionately affected by GDP-tubulin, providing strong new evidence in support of the interface-acting mechanism. Further simulations revealed that nucleotide exchange can modulate the magnitude of plus-end poisoning by GDP-tubulin (*Cleary et al., 2022b*; *Piedra et al., 2016*; *Vandecandelaere et al., 1995*). By ruling out a self-acting (cis-acting) mechanism of nucleotide action, our findings provide new evidence that resolves a longstanding debate about how the bound nucleotide governs the tubulin:tubulin interactions that dictate microtubule growth.

## Results

### Self- and interface-acting mechanisms of nucleotide action predict different effects of GDP-tubulin on plus- and minus-end growth

The self- and interface-acting mechanisms for how nucleotide dictates the strength of tubulin:tubulin interactions are illustrated in *Figure 1A*: the self-acting mechanism posits that the nucleotide bound to β-tubulin (GTP or GDP) controls how tightly *that* tubulin interacts with the lattice, whereas the interface-acting mechanism posits that the nucleotide at the *interface between* tubulin dimers controls how tightly they interact. At the plus-end, the two mechanisms can lead to different outcomes because there are two nucleotides involved – one bound to the terminal tubulin and one at the interface below (*Figure 1A*, top panels). At the minus-end, however, the two mechanisms are indistinguishable because there is only one nucleotide involved: the nucleotide bound to the terminal subunit is also the nucleotide at the interface with the lattice (*Figure 1A*, bottom panels). Using kinetic simulations of microtubule elongation, we sought to identify a testable difference between the self- and interface-acting mechanisms. We first expanded our model (*Ayaz et al., 2014*; *Cleary et al., 2022b*; *Kim and Rice, 2019*; *Piedra et al., 2016*) to simulate both plus- and minus-end elongation and to include multiple nucleotide states for unpolymerized tubulin (summarized in *Figure 1—figure supplement 1*). We then used the model to predict how GDP-tubulin might affect plus- and minus-end elongation with either the self- or interface-acting mechanisms of nucleotide action.

We performed 'mixed nucleotide' simulations of plus- and minus-end growth at 1 μM total tubulin with varying fractions of GDP-tubulin (0–20%). For simplicity, simulations used arbitrarily chosen parameters that supported elongation in the chosen concentration regime (*Table 1*). To provide the simplest possible biochemical setting and to set the stage for experiments described below, simulations did not attempt to explicitly model different conformations of αβ-tubulin (see Discussion), and also ignored GTP hydrolysis. For both self-acting and interface-acting nucleotide, simulated minus-end growth rates decreased identically and in linear proportion to the amount of GDP-tubulin in the simulation (*Figure 1B*). However, simulated plus-end growth rates decreased much more for interface-acting nucleotide than for self-acting nucleotide (*Figure 1C*). Similar results were obtained for alternative parameter choices (*Figure 1—figure supplements 2 and 3*). Based on these robust end-specific differences from simulations, comparative measurements of how GDP-tubulin affects plus- and minus-end elongation should provide a new way to test the self- or interface-acting mechanisms of nucleotide action.

### Mixed nucleotide experiments reveal different effects of GDP-tubulin on plus- and minus-ends

To establish a baseline for measurements with mixed nucleotides, we first used interference reflection microscopy (IRM) to measure plus-and minus-end growth rates in 1 mM GMPCPP (a slowly hydrolyzable GTP analog) at multiple concentrations of bovine brain tubulin. Growth rates displayed the expected linear dependence on tubulin concentration (*Figure 2A*). Both ends showed the same apparent critical concentration ($C_c^{app}$) of 50 nM, but plus-end growth showed a roughly two-fold higher apparent on-rate constant ($k_{on}^{app}$) than minus-end growth, 3 μM$^{-1}$ s$^{-1}$ MT$^{-1}$ and 1.5 μM$^{-1}$ s$^{-1}$ MT$^{-1}$, respectively (*Figure 2B*).

As a way to test the predictions from our simulations (*Figure 1BC*), we next measured the growth rates of microtubule plus- and minus-ends at a constant concentration of tubulin (1.25 μM) but using different ratios of GDP and GMPCPP (1 mM total nucleotide concentration). Growth rates at both ends decreased substantially in mixtures containing as little as 2.5% GDP (25 μM GDP and 975 μM

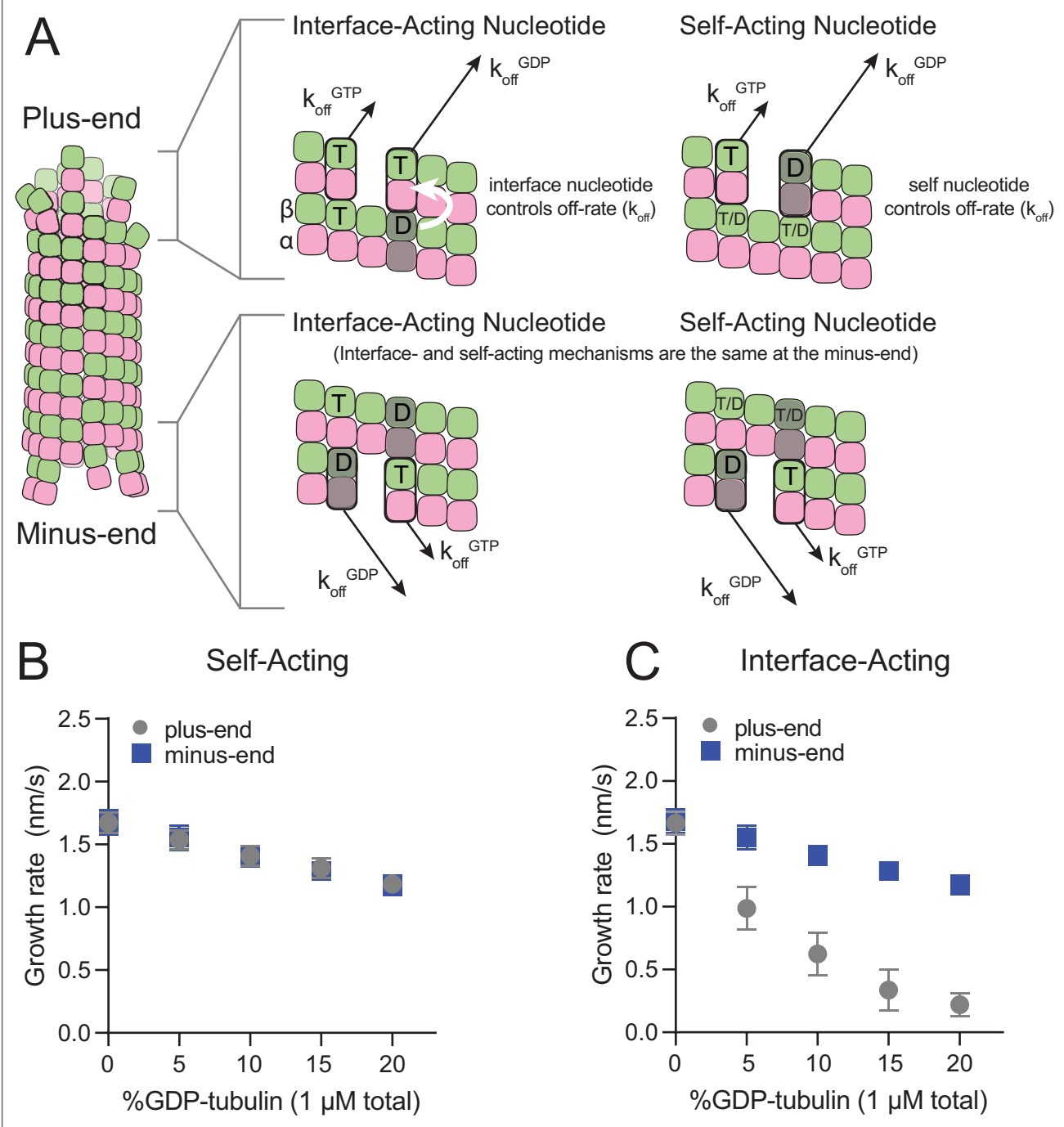

**Figure 1.** Mechanisms of nucleotide action and simulations of plus- and minus-ends. (**A**) Cartoon showing self- (cis) or interface-acting (trans) nucleotide mechanisms. In an interface-acting mechanism, the nucleotide at the interface of two tubulin dimers controls their interaction affinity, shown by a white arrow. In a self-acting mechanism, the nucleotide bound to the terminal tubulin controls how tightly that tubulin interacts with the lattice. At the plus-end, the two mechanisms can lead to different outcomes because there are two nucleotides involved – one bound to the terminal β-tubulin, and one at the interface between the terminal tubulin and the microtubule lattice. At the minus-end, however, self-acting and interface-acting mechanisms are equivalent because the incoming nucleotide becomes the interfacial nucleotide. T=GTP, T/D=GTP or GDP, D=GDP. (**B and C**) Simulated growth rates of GTP microtubule plus- and minus-ends, using arbitrarily chosen parameters that support elongation in the chosen concentration range. (**B**) In a self-acting mechanism, both plus-end (circles) and minus-end (squares) growth rates are predicted to decrease linearly with the amount of GDP-tubulin. (**C**) In an interface-acting mechanism, plus-end (circles) growth rates are predicted to be disproportionately impacted by GDP-tubulin relative to minus-end growth rates. Error bars are standard deviation (n=50 per condition) and if not visible, are obscured by the symbols. Simulation parameters are: $k_{on}$: 1.0 $\mu M^{-1} s^{-1}$, $K_D^{long}$ = 100 $\mu M$, $K_D^{corner}$ = 100 nM, $K_D^{long,GDP}$ = 300 mM. The predicted difference between mechanisms at the plus-end is robust across different

*Figure 1 continued on next page*

*Figure 1 continued*

choices for $K_D^{long}$, $K_D^{corner}$, and the GDP weakening effect (***Figure 1—figure supplement 3***). Note that because the two mechanisms are equivalent at the minus-end, interface-acting simulations for the minus-end use the same simulation results as the self-acting simulations. The total [tubulin] is constant, thus minus-end growth rates decrease in proportion to the decrease in the concentration of GTP-tubulin.

The online version of this article includes the following source data and figure supplement(s) for figure 1:

**Source data 1.** Simulated growth rates for microtubule plus- and minus-ends under different models for nucleotide action.

**Figure supplement 1.** Implementation of plus- and minus-end models.

**Figure supplement 2.** Using simulated growth rates to predict differences between interface- and self-acting nucleotide mechanisms at plus-end and minus-ends.

**Figure supplement 3.** Predicted differences between self- and interface-acting mechanisms at the plus-end are robust to variation in simulation parameters.

GMPCPP) (***Figure 2C***), but plus-end growth rates decreased to a greater degree than minus-end growth rates. For instance, at 25 µM GDP, plus-end growth rates fell ~50% (from 2.2 nm/s to 1.1 nm/s) relative to 'all GMPCPP' growth rates, whereas minus-end growth rates only fell ~30% (from 1 nm/s to 0.7 nm/s). This ~1.5-fold stronger inhibition by GDP of plus-end growth rates held across multiple nucleotide mixing ratios (***Figure 2C***). Importantly, the larger decrease in the growth rate at the plus-end agrees with the predictions made by the interface-acting mechanism (***Figure 1A***) compared to the self-acting mechanism.

## Plus-end growth is super-stoichiometrically suppressed by GDP-tubulin

Tubulin binds different nucleotides with different affinities (***Aldaz et al., 2005***; ***Chakrabarti et al., 2000***; ***Correia et al., 1987***; ***Fishback and Yarbrough, 1984***; ***Hyman et al., 1992***; ***Mejillano and Himes, 1991***; ***Monasterio and Timasheff, 1987***; ***Zeeberg and Caplow, 1979***), so the ratio of GDP and GMPCPP in a reaction does not directly translate to the fractions of GDP- and GMPCPP-tubulin. To estimate the concentrations of GDP- and GMPCPP-tubulin for each nucleotide mixture, we assumed that only GMPCPP-tubulin contributes to minus-ended growth, consistent with our simulations (***Figure 1*** BC). This assumption allowed us to estimate the concentration of GMPCPP-tubulin in each nucleotide mixture by matching the observed growth rates to the control 'all GMPCPP' growth curve (***Figure 2B***). A potential problem with this approach is that the estimated GMPCPP-tubulin concentration in each mixture will be affected by error in the growth rate measurements. To minimize the impact of error, we performed a global fit to all measurements using a simple competitive inhibition model that enforced consistent nucleotide binding affinity (***Figure 3A***). The GMPCPP-tubulin concentrations that best recapitulate minus-end growth rates are consistent with tubulin binding 12.5-fold less tightly to GMPCPP ($K_D^{GMPCPP}$) than to GDP ($K_D^{GDP}$) (***Figure 3B***). This inferred difference of affinities was supported by direct measurements of nucleotide binding (***Figure 3—figure supplement 1***), and agrees with prior reports (***Correia et al., 1987***; ***Hyman et al., 1992***).

To determine whether the observed decrease in growth rate was stoichiometric with the amount of GMPCPP-tubulin in the assay, we used the binding affinities and the known nucleotide concentrations in each mixture (***Figure 3B***) to extrapolate equivalent 'GMPCPP-only' growth rates (***Figure 3*** CD solid lines) from the control 'all GMPCPP' curves (***Figure 2A***). Minus-end growth rates decreased

**Table 1.** Simulation parameters for ***Figures 1–5***.
Dashed lines denote figure panels where either plus-end or minus-end simulations were not performed; these panels focused on simulations of only one end.

| | $k_{on}^{plus}$ (µM⁻¹ s⁻¹) | $k_{on}^{minus}$ (µM⁻¹ s⁻¹) | $K_D^{long}$ (µM) | $K_D^{corner}$ (µM) | $K_D^{long}$ GDP (µM) | $K_D^{corner}$ GDP (µM) | Tubulin (µM) |
|---|---|---|---|---|---|---|---|
| *Figure 1* | 1.0 | 1.0 | 100 | 0.1 | $3\times10^5$ | 300 | 1 |
| *Figure 4A* | 0.74 | 0.31 | 86 | 0.025 | $3\times10^5$ | 87 | 1.25 |
| *Figure 4B* | 0.74 | --- | 86 | 0.025 | $3\times10^5$ | 87 | 1.25 |
| *Figure 4C* | --- | 0.31 | 86 | 0.025 | $3\times10^5$ | 87 | 1.25 |
| *Figure 5B* | 0.74 | --- | 86 | 0.025 | $3\times10^5$ | 87 | 1.25 |

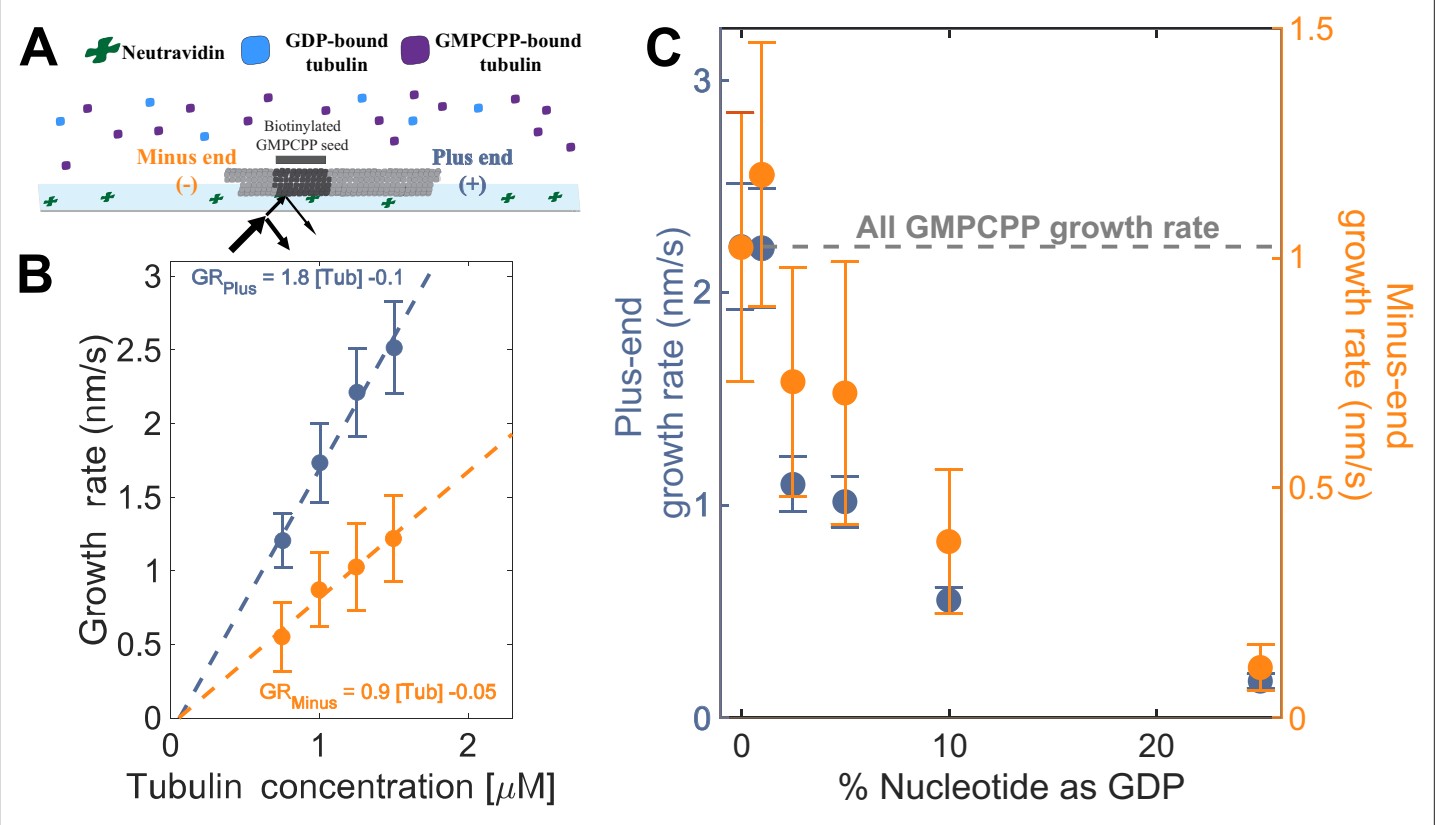

**Figure 2.** Microtubule plus- and minus-end growth both decrease in the presence of GDP-tubulin. (**A**) Schematic of the *in vitro* assay, in which biotinylated GMPCPP microtubule seeds are attached to a neutravidin-coated cover slip, and microtubule assembly in the presence of tubulin bound to either GDP or GMPCPP is monitored using Interference Reflection Microscopy (IRM). (**B**) Growth rates of microtubule plus- and minus-ends in GMPCPP as a function of tubulin concentration (n=64–125 for the plus-end and n=39–95 for the minus-end). The error bars denote standard deviation. (**C**) Plus- (left y-axis) and minus-end (right y-axis) growth rates at 1.25 µM tubulin in mixtures of GDP and GMPCPP containing 1 mM total nucleotide (n=66–121 for the plus-end and n=44–94 for the minus-end). The gray line denotes the 'all GMPCPP' growth rates of the two ends. The error bars denote standard deviation. Using a two-sided t-test with unequal variance, differences in the mean normalized growth rates at plus- and minus-ends were statistically significant with *P*<0.001 for all nucleotide mixtures except 0% GDP.

The online version of this article includes the following source data for figure 2:

**Source data 1.** Measured growth rates for microtubule plus- and minus-ends.

stoichiometrically as the concentration of GDP-tubulin increased, matching or even slightly exceeding the 'all GMPCPP' extrapolation. The only exception was at the highest concentration of GDP-tubulin (*Figure 3C* inset), where growth rates were slow and most challenging to quantify. In contrast, plus-end growth rates decreased super-stoichiometrically (were slower than expected based on the 'all GMPCPP' extrapolation) for a given concentration of GDP-tubulin (*Figure 3D*). This super-stoichiometric effect at the plus-end was observed over a range of GDP-tubulin concentrations, and was most apparent when 25–55% of unpolymerized tubulin was bound to GDP (*Figure 3D* inset). The super-stoichiometric effect of GDP-tubulin on plus-end growth over a wide range of GDP-tubulin concentrations provides strong support for the interface-acting nucleotide mechanism.

## Why is plus-end growth hypersensitive to GDP-tubulin?

To establish a biochemical baseline for simulating mixed nucleotide states, we first fit the interface-acting nucleotide model to the 'all GMPCPP' data (*Figure 2*). The plus-end growth rates were recapitulated well using the same parameters obtained in a prior study (*Cleary et al., 2022b*; $k_{on}^{plus}$ of 0.74 µM$^{-1}$ s$^{-1}$, $K_D^{longitudinal}$ of 86 µM, $K_D^{corner}$ of 25 nM; *Figure 4A*). To extend the model to fit the minus-end growth rates, we retained the same interaction affinities as for the plus-end (consistent with the equal apparent critical concentration at both ends, *Figure 2B*) and optimized a minus-end-specific on-rate constant. This procedure yielded a $k_{on}^{minus}$ of 0.31 µM$^{-1}$ s$^{-1}$ (*Figure 4A*), roughly 2-fold slower than the

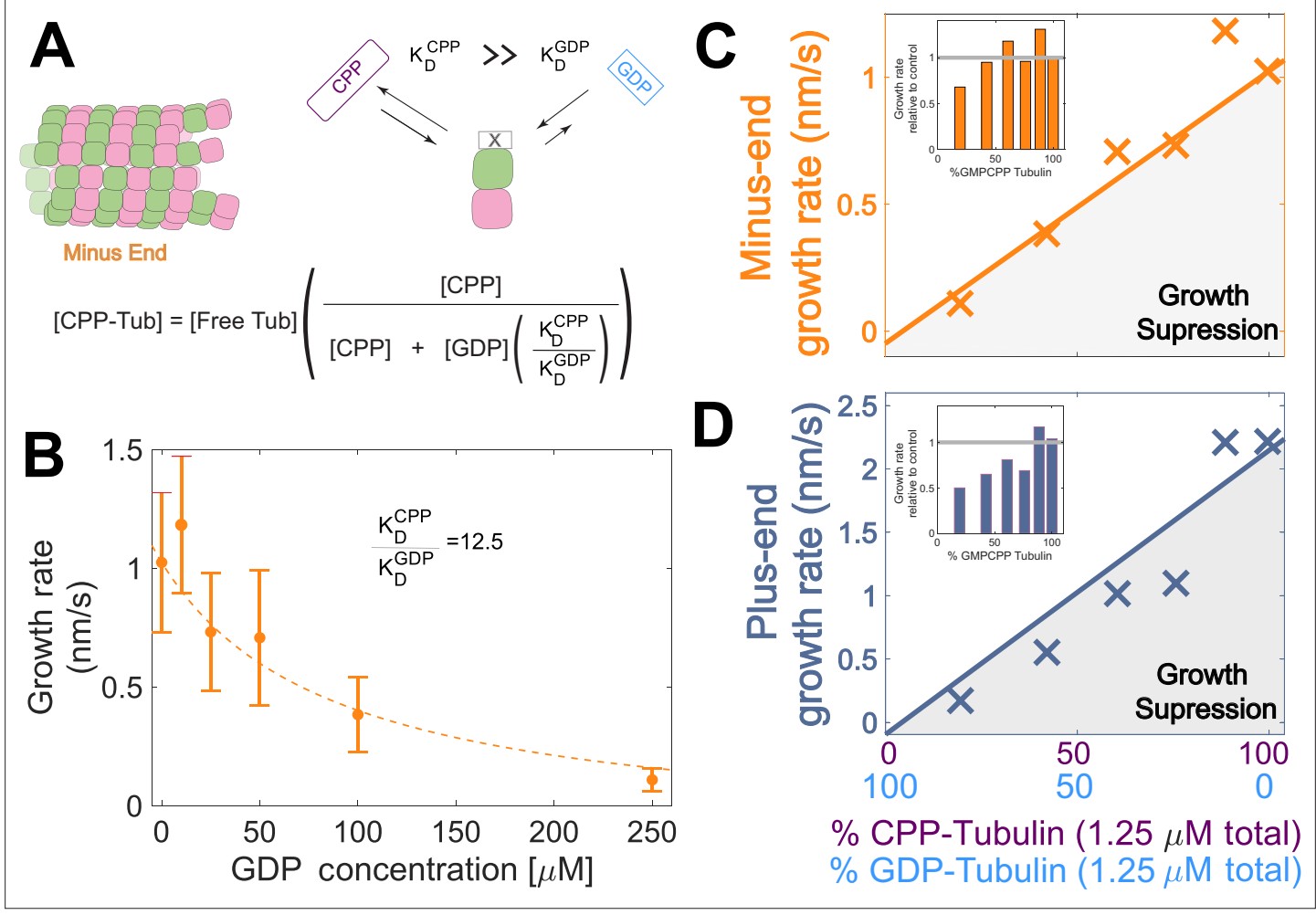

**Figure 3.** Microtubule plus-end growth is suppressed superstoichiometrically by GDP-tubulin. (**A**) Competitive nucleotide binding model. Mixed nucleotide assays result in either GDP- or GMPCPP-bound tubulin landing and creating a nucleotide interface at the minus-end. The concentration of GMPCPP bound tubulin was determined using the concentrations of each nucleotide and their relative affinities ($K_D^{CPP}/K_D^{GDP}$) through a competitive binding model (inset equation). (**B**) Minus-end growth rates as a function of GDP concentration. GMPCPP-tubulin was assumed to be the only tubulin that can contribute to minus-end growth in the mixed nucleotide assays. Minus-end growth rates over varying GDP concentrations were globally fit to a competitive inhibition model (equation in panel A), which resulted in a GMPCPP-tubulin concentration that was consistent with the 'all-GMPCPP' minus-growth curves (*Figure 2B*). The relative affinity of tubulin for GMPCPP compared to GDP ($K_D^{CPP}/K_D^{GDP}$) was the only free parameter in the model. (**C–D**) Growth rates from *Figure 2C* plotted as a function of the fraction of GDP-tubulin, estimated using the known nucleotide content and binding affinities. Growth rates are considered suppressed when falling below the solid lines exhibiting the 'all-GMPCPP' minus- and plus-end growth curves. Insets plot growth rates normalized to the 'GMPCPP-only' growth rates (gray solid line), showing a disproportionate decrease (~1.5-fold for most concentrations) in plus-end growth. Differences in mean normalized growth rates at plus- and minus-ends were statistically significant with *P*<0.001 for all nucleotide mixtures except 0% GDP (see *Figure 2*).

The online version of this article includes the following source data and figure supplement(s) for figure 3:

**Source data 1.** Measured minus-end growth rates as a function of GDP concentration, and plus- and minus-end growth rates plotted vs the concentration of GDP-tubulin.

**Figure supplement 1.** Tubulin has a higher affinity for GDP than for GMPCPP.

plus-end, which is in line with the ~twofold lower concentration-dependence of growth rates observed at the minus-end (*Figure 2A*). We next performed mixed nucleotide (GMPCPP and GDP) simulations at a constant tubulin concentration of 1.25 μM. Simulated minus-end growth rates decreased linearly as the concentration of GDP-tubulin increased, recapitulating the experimental measurements (*Figure 4B*) and in agreement with our initial prediction using arbitrary parameters (*Figure 1*). In contrast, simulated plus-end growth rates decreased super-stoichiometrically as the concentration

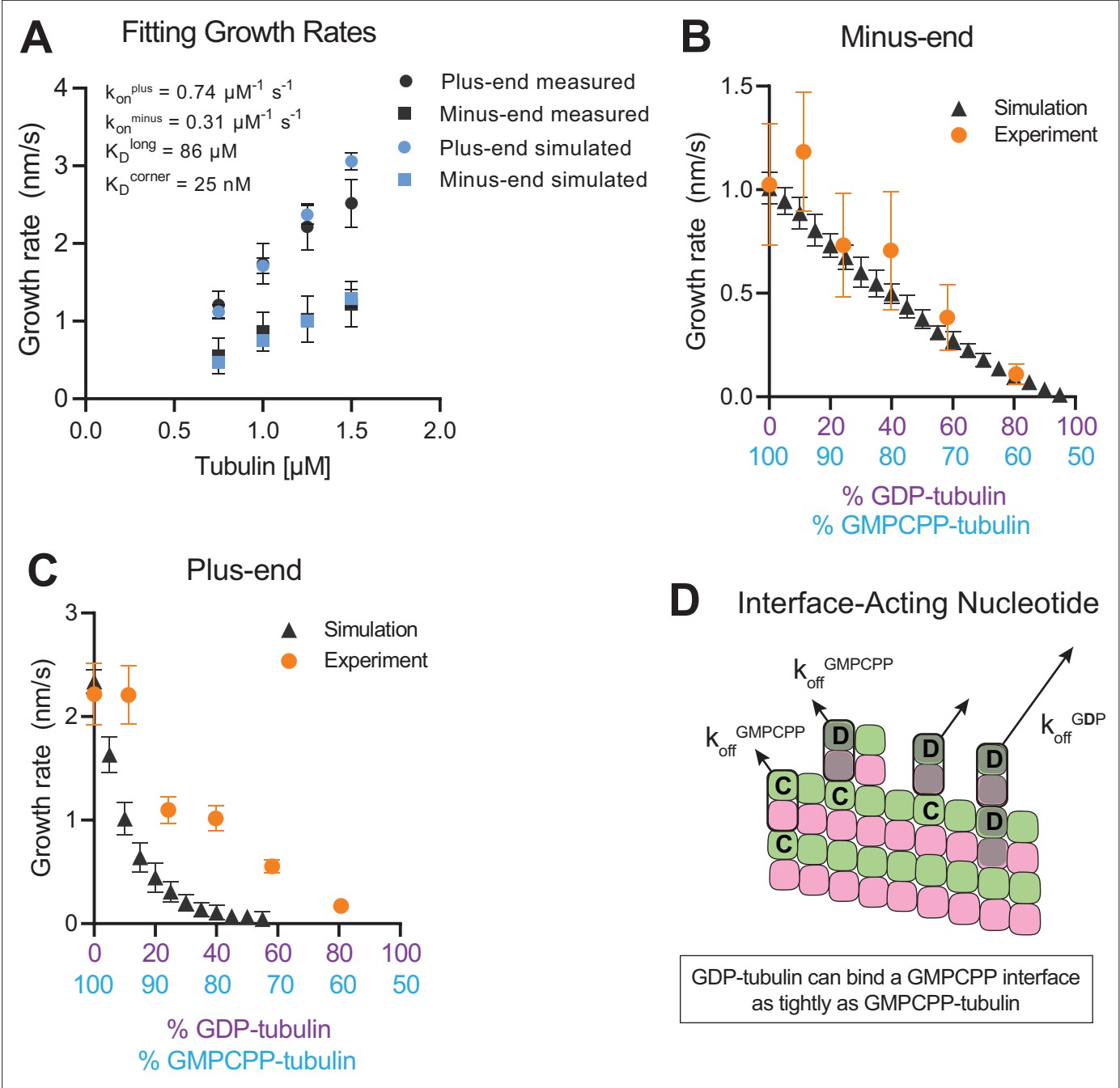

**Figure 4.** Simulating microtubule growth rates in the presence of GDP-tubulin. (**A**) Measured and simulated growth rates for plus- and minus-ends of GMPCPP microtubules. Inset shows the best-fit values for the plus-end and minus-end on-rate constants ($k_{on}^{plus}$ and $k_{on}^{minus}$, respectively), longitudinal interaction ($K_D^{long}$), and corner interaction ($K_D^{corner}$). Error bars show standard deviation (n=50 per simulated concentration) and are obscured by symbols in some cases; experimental data are replotted from *Figure 3*. (**B and C**) Simulated and experimental growth rates at 1.25 µM tubulin in the presence of variable amounts of GDP-tubulin for microtubule minus-ends (**B**) and plus-ends (**C**).(**D**) Cartoon showing how off-rates ($k_{off}$) of GDP-tubulin at the plus-end are dependent upon the interfacial nucleotide; C=GMPCPP, D=GDP (shaded grey). Long-residing GDP-tubulin bound at corner- (one longitudinal and one lateral contact) or bucket-type (one longitudinal and two lateral contacts) binding sites explains the outsized effects of GDP-tubulin on plus-end elongation.

The online version of this article includes the following source data for figure 4:

**Source data 1.** Simulated growth rates after model fitting and how they predict the effect of GDP on plus- and minus-end growth.

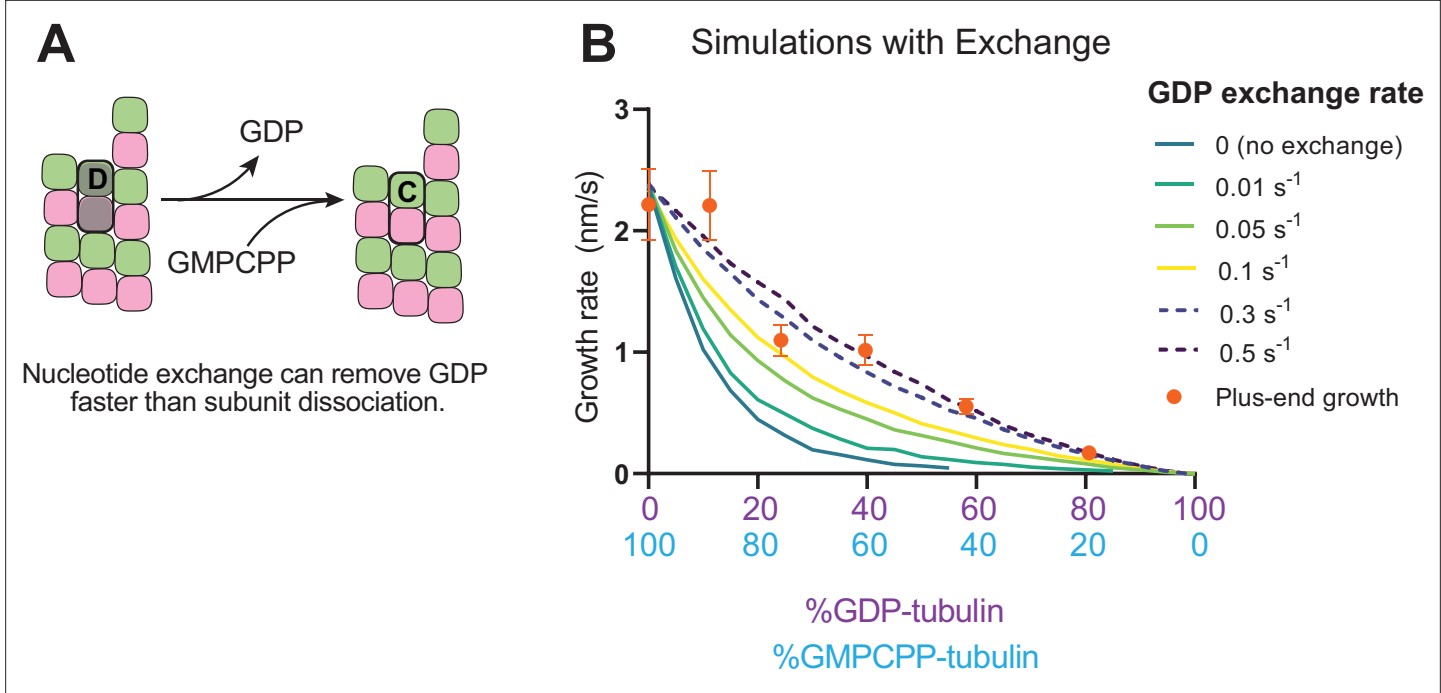

**Figure 5.** Effects of nucleotide exchange on simulated microtubule plus-end growth rates. (**A**) Nucleotide exchange on terminal subunits can mitigate protofilament poisoning at microtubule plus-ends by reducing the lifetime of GDP on the microtubule end. (**B**) Simulated growth rates of microtubule plus-ends as a function of the nucleotide exchange rate (N=50 per simulated concentration, see Methods), showing that faster rates of exchange modulate the effect of protofilament poisoning. Orange circles show the measured plus-end growth rates (*Figure 3*).

The online version of this article includes the following source data and figure supplement(s) for figure 5:

**Source data 1.** How including a finite rate of nucleotide exchange alters predictions of the effect of GDP on growth rate.

**Figure supplement 1.** Implementation and analysis of nucleotide exchange.

of GDP-tubulin increased (*Figure 4C*). An outsized effect of GDP-tubulin at the plus-end is expected from the interface-acting mechanism, but the model overpredicted the magnitude of the effect.

In the interface-acting nucleotide mechanism, the outsized effect of GDP at the plus-end occurs because GDP-tubulin can bind tightly to (and reside longer at) the plus-end if the interfacial nucleotide is GMPCPP (*Figure 4D*). This 'extended stay' of GDP-tubulin on the plus-end poisons the protofilament end against further growth and is the origin of the super-stoichiometric effect of GDP-tubulin at the plus-end (*Figure 4D*). We reasoned that our model might be overpredicting the magnitude of the GDP-poisoning effect (*Figure 4C*) because it was neglecting some other mechanism that normally limits the lifetime of GDP-tubulin at the plus-end.

### Nucleotide exchange at the plus-end can alleviate protofilament 'poisoning' by GDP-tubulin

Recent work (*Luo et al., 2023*; *Piedra et al., 2016*) has reinforced early results (*Chen and Hill, 1983*; *Chen and Hill, 1985*; *Mitchison, 1993*) that pointed to the potential role of nucleotide exchange in microtubule dynamics at the plus-end. We implemented a finite rate of nucleotide exchange in the model (see Methods) to determine whether exchange might allow the simulations to better recapitulate the magnitude by which GDP-tubulin super-stoichiometrically decreased plus-end growth (*Figure 5A*). We performed interface-acting plus-end simulations using a range of nucleotide exchange rates (*Figure 5—figure supplement 1*). Faster rates of nucleotide exchange yielded smaller decreases in plus-end growth rates for a given concentration of GDP-tubulin (*Figure 5B*). The rate of nucleotide exchange that best recapitulated the observed effects was 0.3–0.5 s⁻¹, which compares favorably to other estimates (*Amayed et al., 2000*; *Melki et al., 1989*; *Yarbrough and Fishback, 1985*; *Figure 5—figure supplement 1*). In summary, using simulations and measurements of plus- and

minus-end growth, we showed that microtubule plus- and minus-ends exhibit different sensitivities to GDP-tubulin, lending strong support for the interface-acting mechanism of nucleotide action.

## Discussion

A connection between tubulin nucleotide state and microtubule stability has long been appreciated, but the molecular mechanism underlying the connection has been surprisingly difficult to determine. At one extreme, a self-acting mechanism inspired by conformational differences between unpolymerized and polymerized tubulin posits that GTP dictates microtubule stability by promoting a more microtubule-compatible conformation for the tubulin *to which it is bound*. At the other extreme, an interface-acting mechanism inspired by direct participation of nucleotide in tubulin:tubulin polymerization contacts posits that the nucleotide influences the behavior of the *next* tubulin along the protofilament. Ruling out either the self- or interface-acting mechanism has been challenging because it has not been possible to manipulate the nucleotide on the plus-end separately from the nucleotide on unpolymerized tubulin. Consequently, tests to date have relied on indirect data.

In the present study, we took advantage of microtubule polarity to address the debate about the mechanism of nucleotide action in a new way. Our approach rests on an asymmetry in the way that nucleotide participates in plus- and minus-end interactions. At the minus-end, there is no difference between self- and interface-acting nucleotide mechanisms because the nucleotide on the terminal tubulin is also the interfacial nucleotide that participates in contacts along the protofilament. At the plus-end, however, two different nucleotide binding sites are involved: the one exposed on the terminal tubulin, and one at the interface with (underneath) the terminal tubulin. Our computational simulations of microtubule elongation revealed that the two mechanisms make different predictions about the sensitivity of plus- and minus-end elongation to GDP-tubulin. We used mixed nucleotide (GMPCPP and GDP) experiments to measure the effects of GDP-tubulin on elongation of plus- and minus-ends in a way that controls nucleotide state(s) while also avoiding complications associated with microtubule catastrophe.

We observed that the decrease in elongation rate was proportionally greater for the plus-end than for the minus-end over a wide range of GDP-tubulin fractions. This outsized effect at the plus-end is consistent with an earlier study that observed a loss of plus-end elongation when using mixtures of GTP and GDP (*Tanaka-Takiguchi et al., 1998*). The outsized effect at the plus-end conforms to predictions of the interface-acting mechanism and is incompatible with the self-acting mechanism. To recapitulate the magnitude of GDP-tubulin-induced suppression of growth rates, the simulations required a finite rate of nucleotide exchange on plus-end protofilaments. Our experiments and simulations provide strong new data that support the interface-acting mechanism of nucleotide action.

The outsized effect of GDP on plus-ends provides new insights into the fundamental mechanisms of microtubule dynamics and adds to a growing body of evidence that suggests GDP-terminated protofilaments influence microtubule growth (*Carlier and Pantaloni, 1978*; *Hamel et al., 1986*; *Valiron et al., 2010*), fluctuations (*Cleary et al., 2022b*), catastrophe (*Caplow and Shanks, 1996*; *Piedra et al., 2016*), and regulation (*Lawrence et al., 2022*; *Luo et al., 2023*). Our results point to a more nuanced view of the GTP cap model, which posits that growing microtubule ends are protected against depolymerization by a 'cap' of GTP-tubulin (*Mitchison and Kirschner, 1984*), reviewed in *Gudimchuk and McIntosh, 2021*. Early views of the GTP cap did not anticipate the influence of GDP-tubulin on growing plus-ends, but there is now increasing evidence (*Carlier and Pantaloni, 1978*; *Farmer and Zanic, 2023*; *Hamel et al., 1986*; *Margolin et al., 2012*; *Maurer et al., 2012*; *Roth et al., 2018*; *Valiron et al., 2010*) that the cap is not 'all or nothing', and that GDP-tubulin can modulate microtubule growth without always initiating a catastrophe. Indeed, the tendency for plus-ended growth to 'stutter' (*Mahserejian et al., 2022*) and fluctuate (*Cleary et al., 2022b*) might be explained by exposed GDP-tubulin; exposed GDP-tubulin may also contribute to the higher frequency of catastrophe at the plus-end (*Strothman et al., 2019*; *Walker et al., 1988*). Our work supports an emerging view of the growing microtubule end as a 'mosaic' of nucleotide states rather than a uniform assembly of GTP-tubulin (*Brouhard and Sept, 2012*; *Brouhard and Rice, 2018*; *Cross, 2019*; *Duellberg et al., 2016*; *Farmer et al., 2021*; *Farmer and Zanic, 2023*; *Gudimchuk and McIntosh, 2021*; *Howard and Hyman, 2009*; *Margolin et al., 2012*; *Maurer et al., 2012*; *Roostalu et al., 2020*; *Roth et al., 2018*). By allowing for the possibility of multiple nucleotide states on the microtubule end, our work also resonates with recent studies of the microtubule regulatory factor CLASP (*Lawrence et al.,*

*2022*; *Luo et al., 2023*), which regulates microtubule plus-ends differently depending on the nucleotide state of the terminal subunit at the protofilament plus-end.

Our modeling purposefully implemented the simplest forms of self- and interface-acting nucleotide mechanisms. We did not attempt to explicitly model how conformations of αβ-tubulin might influence the strength of tubulin:tubulin interactions: there is no consensus about how to do so, and modeling different conformations introduces substantially more adjustable parameters into the model, which complicates fitting and interpretation (*Coombes et al., 2013*; *Molodtsov et al., 2005*; *Stewman et al., 2020*; *VanBuren et al., 2005*; *Zakharov et al., 2015*). In support of a simpler model, our use of GMPCPP and GDP mixtures simplified the biochemical picture by ensuring that, except for the very end, the microtubule lattice will be predominantly in a single nucleotide state (GMPCPP). This choice diminishes the importance of explicitly modeling different conformations. The model might also implicitly capture a subset of the conformation-dependent effects on tubulin:tubulin interfaces because the longitudinal and corner affinities are refined independently: the strength of longitudinal interactions could therefore be different for corner than for pure longitudinal sites, potentially reflecting the cost of tubulin 'straightening' during polymerization. In the interest of minimizing the number of adjustable parameters in the model, we also did not consider 'hybrid' models incorporating elements from both self- and interface-acting mechanisms. While we acknowledge the possibility that self-acting mechanisms may contribute to modulation of plus-end stability, the large differences we predicted and observed between plus- and minus-ends indicate that interface-acting nucleotide effects are sufficient to explain the observations. This interface-centric view of nucleotide action is also consistent with cryo-EM studies, which show that the largest nucleotide-dependent conformational changes in the microtubule occur in the α-tubulin subunit above and directly contacting the β-tubulin exchangeable nucleotide (*Alushin et al., 2014*; *Manka and Moores, 2018*; *Zhang et al., 2015*).

In summary, the findings reported here provide the most direct evidence to date in support of an interface-acting mechanism for nucleotide in microtubule stabilization. Depolymerizing microtubules can also perform mechanical work, and it is interesting to consider parallels with other work-performing, oligomeric nucleotide hydrolases. The curling protofilaments that occur during microtubule depolymerization are effectively linear oligomers held together (and to the microtubule end) by nucleotide-dependent interactions at the tubulin:tubulin interfaces. AAA-family proteins, which are oligomeric ATPases that use nucleotide-dependent reorganization of quaternary structure to unfold proteins, package viral DNA, and remodel the structure of nucleic acids (*Banerjee et al., 2016*; *Brunger and DeLaBarre, 2003*; *Davies et al., 2005*; *Erzberger and Berger, 2006*), also appear to use an interface-acting mechanism for their bound adenosine nucleotide. Indeed, just as the GTP-binding site on β-tubulin forms part of the longitudinal interface between tubulin subunits, the ATP binding site in AAA proteins resides at a protomer:protomer interface and dictates the geometry of oligomerization contacts (*Erzberger and Berger, 2006*). Furthermore, for both AAA proteins and microtubules, residues important for nucleotide hydrolysis on one subunit are contributed by the next subunit in the oligomer or polymer (*Banerjee et al., 2016*; *Brunger and DeLaBarre, 2003*; *Davies et al., 2005*). We speculate that these similarities involving interfacial nucleotides in otherwise unrelated proteins may indicate a shared, convergently evolved mechanism for achieving force production in oligomers.

## Methods
### Protein purification and labeling

PC-grade bovine brain tubulin was purified as previously described (*Cleary et al., 2022b*; *Uppalapati et al., 2009*), double cycled, quantified by absorbance at 280 nm ($\varepsilon_{tubulin}$ of 115,000 M$^{-1}$ cm$^{-1}$), diluted to 100 µM in BRB80 (80 mM K-Pipes, 2 mM EGTA, 2 mM MgCl$_2$, pH 6.9), aliquoted, flash frozen in liquid nitrogen, and stored at –80 °C. Prior to experiments, tubulin aliquots were thawed on ice, diluted to 20 µM in BRB80, and concentrations reconfirmed by A$_{280}$.

Tubulin was biotinylated as previously described (*Cleary et al., 2022b*). Briefly, microtubules were polymerized by combining 40 µM tubulin, 1 mM GTP, 1 mM MgCl$_2$ and 5% DMSO in BRB80, incubating at 37 °C for 30 min. An equimolar amount of EZ-Link NHS-Biotin in DMSO (ThermoFisher 20217) was added and allowed to react for 30 min at 37 °C. Microtubules were then pelleted, the pellet resuspended in cold BRB80 and incubated on ice for 30 min to depolymerize the microtubules, the solution centrifuged at 30 psi for 10 min in a Beckman Airfuge using a pre-chilled rotor, and

supernatant collected. This biotinylated tubulin was then cycled, the tubulin concentration checked by $A_{280}$, and the degree of biotinylation quantified using the Biocytin Biotin Quantification Kit (Thermo Fisher Scientific #44610). Final stocks of biotinylated tubulin were mixed with unlabeled tubulin to 40 µM total tubulin to obtain a 33% biotin-labeled fraction, aliquoted, frozen in liquid nitrogen, and stored at –80 °C.

Biotinylated microtubule seeds were polymerized by combining 20 µM biotinylated tubulin (33% biotin-labeled), 1 mM GMPCPP (Jena Biosciences) and 4 mM $MgCl_2$, and incubating at 37 °C for 1 hr. The seeds were then elongated by diluting the total tubulin concentration to 2 µM in BRB80 with 0.5 mM GMPCPP and 2 mM $MgCl_2$ and incubating for 5 hr at 37 °C. The seeds were pelleted, resuspended in BRB80 with 20% glycerol, flash frozen in liquid nitrogen, and stored at –80 °C. On the day of experiments, the aliquot was rapidly thawed at 37 °C, the seeds pelleted to remove glycerol, and resuspended in a solution containing 0.5 mM Mg-GMPCPP.

## Microtubule dynamics assays

Coverslips (18×18 mm Corning) were cleaned in 7X Cleaning Detergent (MP Biomedicals 097667093) diluted to 1X in $ddH_2O$. The solution was heated at 45 °C until clear, the coverslips were then immersed for 2 hours, removed and rinsed with $ddH_2O$, and plasma cleaned (Harrick Plasma) for 12 min. Following cleaning, coverslips were silanized by incubating in a vacuum-sealed desiccator with 1 H,1H,2H,2H-perfluorodecyltrichlorosilane (Alfa Aesar L165804-03) overnight. Before use, the degree of silanization was checked using a droplet test to confirm hydrophobicity.

To construct flow cells, a second ethanol-washed and $ddH_2O$-rinsed coverslip (60x24 mm Corning) was scored, split to a width smaller than 18 mm, and attached to the silanized coverslip with two strips of double-sided tape spaced roughly 10 mm apart. For the experiment, 600 nM neutravidin (Thermo Fisher) was flowed into the chamber, followed by 5% F127 (Sigma P2443-250G), 2 mg/mL casein (Sigma C-7078), and biotinylated microtubule seeds at a concentration that resulted in approximately 10 seeds per 90x90 µm² field of view. Biotinylated BSA (1 mg/mL) was then added to the flow chamber to block any free neutravidin on the cover slip.

Due to the slow growth conditions in these experiments, it was necessary to pre-establish the plus- and minus-ends of the seeds in every field of view. Polarity was determined by injecting into the flow cell a solution containing 12.5 µM tubulin, 1 mM Mg-GTP and an oxygen scavenging system consisting of 80 µg/mL Catalase (Sigma C1345-1G), 100 mM DTT, 200 mM D-Glucose (EMD Millipore Corp DX0145-1), and 200 µg/mL Glucose Oxidase (EMD Millipore Corp 345386–10 gm) in BRB80. The flow cell was allowed to warm for 5 min in contact with the objective of the Nikon TE-2000 TIRF with an objective heater set to 30 °C. Microtubules were visualized using IRM with a blue (440 nm) LED at 0.5% power (pE-300white, CoolLED, UK). Microtubule growth in GTP was monitored for 5 min, and the faster growing end of each microtubule in the field was defined as the plus-end. The tubulin solution was then replaced with tubulin-free cold BRB80, the flow cell was incubated for 5 min to depolymerize the microtubules with depolymerization confirmed by visualization, and finally any residual tubulin was removed by flowing through five flow cell volumes of cold BRB80.

While monitoring the same field of view, a solution was introduced containing tubulin, 1 mM nucleotide (either Mg-GMPCPP or a mixture of Mg-GDP/GMPCPP, with concentrations quantified by absorbance at 252 nm, using $\varepsilon$=13,700 M⁻¹ cm⁻¹), and an oxygen scavenging system. Once the final polymerization mixture was introduced, the chamber was sealed with nail polish and allowed to equilibrate to 30 °C while in contact with the objective. Images were subsequently taken at 1 frame per second for up to 2.5 hr. All measurements were performed at least two separate times, except for the 25% GDP condition.

## Image analysis and processing

Each video was flat-fielded to correct for uneven illumination, as follows using ImageJ (*Schindelin et al., 2015*). First, an out-of-focus movie was acquired and a median image generated. The median image was then converted to 32-bits, and normalized to 1 by dividing every pixel value by the mean pixel intensity in the image. Finally, every experimental video was flat-fielded by dividing the intensity values in every frame by this normalized median image. Stage drift was corrected as previously described (*Cleary et al., 2022b*): fiducial markers were tracked using FIESTA (*Ruhnow et al., 2011*) and used as input for an in-house drift correction program written in Matlab. To quantify microtubule

growth, kymographs were generated from pixel-corrected movies using the line-scan tool in ImageJ. Plus- and minus-end growth rates were determined by fitting a line to smooth and continuous growth events and calculating the slope.

## Global fit of minus-end growth

The relative binding affinities of tubulin for GDP and GMPCPP were estimated by fitting the minus-end growth rates at varying nucleotide ratios to a model in which only GMPCPP-tubulin contributes to minus-end growth, as follows. From **Figure 2B**, the minus-end growth rate (GR$_{minus}$ in nm/s) as a function of [tubulin$_{GMPCPP}$] (in µM) was:

$$GR_{minus} = 0.9 * \left[tubulin_{GMPCPP}\right] - 0.05 \tag{1}$$

In a mixture of GMPCPP and GDP, the concentration of GMPCPP-tubulin can be determined by a competitive binding model (analogous to competitive inhibition of an enzyme **Cheng and Prusoff, 1973**) in which the two nucleotides compete for binding to tubulin:

$$\left[tubulin_{GMPCPP}\right] = \left[tubulin_{total}\right] \left( \frac{\left[GMPCPP\right]}{\left[GMPCPP\right] + K_D^{GMPCPP} \left(1 + \frac{\left[GDP\right]}{K_D^{GDP}}\right)} \right) \tag{2}$$

Because [GMPCPP] was relatively high in all cases, we made the assumption that $\left[GMPCPP\right] \gg K_D^{GMPCPP}$, which simplifies **Equation 2** to:

$$\left[tubulin_{GMPCPP}\right] = \left[tubulin_{total}\right] \left( \frac{\left[GMPCPP\right]}{\left[GMPCPP\right] + \left[GDP\right] \frac{K_D^{GMPCPP}}{K_D^{GDP}}} \right) \tag{3}$$

Plugging (3) into (1) gives:

$$GR_{minus} = 0.9 * \left[tubulin_{total}\right] \left( \frac{\left[GMPCPP\right]}{\left[GMPCPP\right] + \left[GDP\right] \frac{K_D^{GMPCPP}}{K_D^{GDP}}} \right) - 0.05 \tag{4}$$

Finally, because the total nucleotide concentration was kept constant at 1000 µM, we could replace [GMPCPP] by 1000 − [GDP], yielding:

$$GR_{minus} = 0.9 * \left[tubulin_{total}\right] \left( \frac{1000 - \left[GDP\right]}{1000 - \left[GDP\right] + \left[GDP\right] \frac{K_D^{GMPCPP}}{K_D^{GDP}}} \right) - 0.05 \tag{5}$$

The minus-end growth rates as a function of [GDP] in **Figure 2B** (where $\left[tubulin_{total}\right]$ was 1.25 µM) were fit to **Equation 5**. Here, the only free parameter is the relative affinity of tubulin for GMPCPP and GDP ($K_D^{GMPCPP}$ / $K_D^{GDP}$). The fit was weighted by the inverse of the standard error of the mean (SEM).

## Nucleotide binding affinity assays

The affinity of tubulin for GMPCPP and GDP was determined using a competition assay that relies on the quenching of tryptophan fluorescence by 6-Thio GTP (**Amayed et al., 2000**; **Fishback and Yarbrough, 1984**; **Piedra et al., 2016**). Aliquots of tubulin (~80 µM) and Bovine Serum Albumin (BSA – 50 mg/mL) were rapidly thawed, filtered through a 0.1 µm spin filter (Millipore-Sigma, UFC30VV25) at 11,000 rpm and 4 °C to remove aggregates, and concentrations quantified by absorbance at A$_{280}$ with ε$_{tubulin}$=115,000 M$^{-1}$ cm$^{-1}$ and ε$_{BSA}$=43,824 M$^{-1}$ cm$^{-1}$. The affinity of 6-Thio GTP for tubulin was measured by preparing 220 µL samples of either 0.2 µM tubulin or 0.56 µM BSA with varying concentrations

of 6-Thio GTP. The BSA concentration was chosen to match the tryptophan fluorescence of tubulin, which allowed for the correction of the inner filter effect due to absorbance of 6-Thio GTP at the tryptophan emission peak. A buffer-only well was included in every plate as a zero fluorescence control, and the value of the blank was subtracted from each BSA and tubulin measurement. Tryptophan fluorescence readings (297 nm excitation and 332 nm emission) were performed in 96-well, flat bottom, UV-star plates (Greiner bio-one, 655809) on a Molecular Devices FlexStation 3 Multimode Microplate Reader. Each recorded fluorescence value was an average of 250 signal determinations. The fluorescence readings were corrected for the inner filter effect by dividing the tubulin fluorescence signal by the BSA fluorescence signal at each nucleotide concentration (*Fishback and Yarbrough, 1984*). The standard error of the mean (SEM) was calculated using propagated errors of the relative SEM for each variable:

$$SEM = \left( \sqrt{\frac{\left(SEM_{BSA}\right)^2}{\bar{BSA}} + \frac{\left(SEM_{Tubulin}\right)^2}{\bar{Tubulin}} + \frac{\left(SEM_{Blank}\right)^2}{\bar{Blank}}} \right) * \left( \frac{\bar{Tubulin}}{\bar{BSA}} \right)$$

The affinity of tubulin for 6-Thio GTP ($K_D^{6\text{-Thio GTP}}$) was determined by adding increasing concentrations of 6-Thio GTP, measuring the fall in fluorescence due to fluorescence quenching by the nucleotide, and fitting the data to a binding isotherm weighted by the inverse of the SEM:

$$y = B - \left( \frac{A * \left[6 - ThioGTP\right]}{\left[6 - ThioGTP\right] + K_D^{6-ThioGTP}} \right)$$

where A corresponds to the amplitude of the fall in fluorescence and (B - A) is the remaining fluorescence under full quenching conditions. Competition assays were then performed by adding increasing concentrations of GMPCPP or GDP to a solution containing 3 µM 6-Thio GTP. Competition between the unlabeled nucleotides and the 6-Thio GTP caused unquenching of fluorescence, allowing for determination of the affinity of tubulin for GMPCPP ($K_D^{GMPCPP}$) and GDP ($K_D^{GDP}$). Data for each nucleotide were fit to a competition model:

$$y = C + \left( \frac{A * \left[Nucleotide\right]}{\left[Nucleotide\right] + K_D^{Nucleotide}\left(1 + \frac{\left[6 - ThioGTP\right]}{K_D^{6-ThioGTP}}\right)} \right)$$

Here, A is the amplitude fluorescence quenching, which was constrained by the measured value at 3 µM 6-Thio GTP, and C is a free parameter corresponding to the quenched fluorescence value at zero unlabeled nucleotide. For this fit, the means were weighted by the inverse of the SEM.

## Simulating microtubule growth of plus- and minus-ends

Simulations of plus- and minus-end elongation were performed using extended versions of previously-described code and analysis algorithms (*Cleary et al., 2022a*; *Cleary et al., 2022b*; *Kim and Rice, 2019*; *Mickolajczyk et al., 2019*; *Piedra et al., 2016*). The main features of the model are outlined in *Figure 1—figure supplement 1* and described in detail in our previous publication (*Cleary et al., 2022b*). Briefly, the code performs kinetic Monte Carlo simulations of microtubule elongation at the level of individual association and dissociation events, creating a 'biochemical movie' of polymerization with one reaction (association, dissociation, or nucleotide exchange) per frame. First-order subunit association rate constants are calculated by multiplying the bimolecular on-rate constant ($k_{on}$) by the tubulin concentration (on-rate=$k_{on}$*[tubulin]). Subunit dissociation rates ($k_{off} = k_{on}$*$K_D$) are dependent on the interaction affinity ($K_D$) at a specific site, which is determined by the number and type of tubulin-tubulin interactions at the respective site.

The simulation code from the present work is available as a GitLab repository (*Rice et al., 2023*). To compare the effects of interface-acting and self-acting mechanisms, plus-end simulation code was modified to implement self-acting (cis) nucleotide instead of interface-acting nucleotide (*Figure 1—figure supplement 1B*). To simulate the minus-end, simulation rules were updated to reflect the lack of an exposed nucleotide on the minus-end, and the orientation of interactions across the seam (*Figure 1—figure supplement 1C*). Simulating GDP- and GMPCPP-tubulin mixtures required two

**Table 2.** Calculated on-rates and off-rates for simulations presented in *Figures 1–5*.
On-rate is calculated using the biochemical $k_{on}$ and the concentration of tubulin [µM]. Off-rate is calculated using the biochemical $k_{on}$ and the dissociation constant $K_D$.

| | Simulated end | On-rate (s⁻¹) | $k_{off}^{long}$ (s⁻¹) | $k_{off}^{corner}$ (s⁻¹) | $k_{off}^{long}$ GDP (s⁻¹) | $k_{off}^{corner}$ GDP (s⁻¹) | Tubulin (µM) |
|---|---|---|---|---|---|---|---|
| *Figure 1* | plus | 1 | 100 | 0.1 | $3\times10^5$ | 300 | 1 |
| | minus | 1 | 100 | 0.1 | $3\times10^5$ | 300 | 1 |
| *Figure 4A* | plus | 0.9 | 64 | 0.019 | $2.2\times10^5$ | 65 | 1.25 |
| | minus | 0.4 | 27 | 0.0078 | $9.3\times10^4$ | 27.3 | 1.25 |
| *Figure 4B* | plus | 0.9 | 64 | 0.019 | $2.2\times10^5$ | 65 | 1.25 |
| *Figure 4C* | minus | 0.4 | 27 | 0.0078 | $9.3\times10^4$ | 27.3 | 1.25 |
| *Figure 5B* | plus | 0.9 | 64 | 0.019 | $2.2\times10^5$ | 65 | 1.25 |

new parameters: (1) the concentration of GDP-tubulin and (2) a factor to weaken GDP-mediated contacts, represented as a multiplicative factor on the GMPCPP interaction affinity. We assumed that there were no inherent differences between the association rates of GMPCPP- and GDP-tubulin, and used the same on-rate constant ($k_{on}^{plus}$ or $k_{on}^{minus}$, respectively) for GDP- and GMPCPP-tubulin.

We generalized our prior implementation of nucleotide exchange (*Piedra et al., 2016*) to allow all terminal nucleotides (whether GMPCPP or GDP) to exchange. The probability of replacement by GDP or GMPCPP was set to be proportional to the fractional concentration of either nucleotide. Our implementation assumes that the rate-limiting step in nucleotide exchange is dissociation of the (previously) bound nucleotide. To reflect the 12.5-fold difference between the affinity of tubulin for GDP and GMPCPP, the rate of GMPCPP exchange (the off-rate) was set to be 12.5-fold faster than the rate of GDP exchange (as measured in *Figure 3*).

**Table 3.** Simulation parameters for *Figure 1—figure supplement 1*.
Shading has been added to highlight which simulation parameters were changed, with respect to the reference parameters used in *Figure 1*. The GDP fold weaker values are the fold change between the GTP- and GDP-type interaction.

| Figure 1—figure supplement 3 | change | $k_{on}$ (µM⁻¹ s⁻¹) | $K_D^{long}$ GTP (µM) | $K_D^{corner}$ GTP (µM) | $K_D^{long}$ GDP (µM) | $K_D^{corner}$ GDP (µM) | GDP$^{long}$ fold weaker | GDP$^{corner}$ fold weaker | Tubulin (µM) |
|---|---|---|---|---|---|---|---|---|---|
| *Figure 1—figure supplement 3A* | GDP weakening | 1.0 | 100 | 0.1 | $3\times10^6$ | 3000 | 30,000 | 30,000 | 1 |
| *Figure 1—figure supplement 3A* | GDP weakening | 1.0 | 100 | 0.1 | $3\times10^5$ | 300 | 3,000 | 3,000 | 1 |
| *Figure 1—figure supplement 3* | GDP weakening | 1.0 | 100 | 0.1 | $3\times10^4$ | 30 | 300 | 300 | 1 |
| *Figure 1—figure supplement 3* | $K_D^{long}$ | 1.0 | 1000 | 0.1 | $3\times10^6$ | 300 | 3000 | 3000 | 1 |
| *Figure 1—figure supplement 3* | $K_D^{long}$ | 1.0 | 100 | 0.1 | $3\times10^5$ | 300 | 3000 | 3000 | 1 |
| *Figure 1—figure supplement 3* | $K_D^{long}$ | 1.0 | 10 | 0.1 | $3\times10^4$ | 300 | 3000 | 3000 | 1 |
| *Figure 1—figure supplement 3* | $K_D^{corner}$ | 1.0 | 100 | 0.5 | $3\times10^5$ | 1500 | 3000 | 3000 | 1 |
| *Figure 1—figure supplement 3* | $K_D^{corner}$ | 1.0 | 100 | 0.1 | $3\times10^5$ | 300 | 3000 | 3000 | 1 |
| *Figure 1—figure supplement 3* | $K_D^{corner}$ | 1.0 | 100 | 0.02 | $3\times10^5$ | 60 | 3000 | 3000 | 1 |

**Table 4.** Calculated on-rates and off-rates for simulations in *Figure 1—figure supplement 1*. On-rate is calculated using the biochemical $k_{on}$ and the concentration of tubulin [µM]. Off-rate is calculated using the biochemical $k_{on}$ and the dissociation constant $K_D$. All simulations in *Figure 1—figure supplements 2 and 3* use the same biochemical $k_{on}$ for plus-end and minus-end simulations, as was done in *Figure 1*. Shading highlights which off-rates changed, with respect to the original values in *Figure 1*.

| Figure 1—figure supplement 3 | On-rate (s⁻¹) | $k_{off}^{long}$ GTP (s⁻¹) | $k_{off}^{corner}$ GTP (s⁻¹) | $k_{off}^{long}$ GDP (s⁻¹) | $k_{off}^{corner}$ GDP (s⁻¹) | Tubulin (µM) |
|---|---|---|---|---|---|---|
| *Figure 1—figure supplement 3A* | 1 | 100 | 0.1 | $3\times10^6$ | 3000 | 1 |
| *Figure 1—figure supplement 3A* | 1 | 100 | 0.1 | $3\times10^5$ | 300 | 1 |
| *Figure 1—figure supplement 3A* | 1 | 100 | 0.1 | $3\times10^4$ | 30 | 1 |
| *Figure 1—figure supplement 3B* | 1 | 1000 | 0.1 | $3\times10^6$ | 300 | 1 |
| *Figure 1—figure supplement 3B* | 1 | 100 | 0.1 | $3\times10^5$ | 300 | 1 |
| *Figure 1—figure supplement 3B* | 1 | 10 | 0.1 | $3\times10^4$ | 300 | 1 |
| *Figure 1—figure supplement 3C* | 1 | 100 | 0.5 | $3\times10^5$ | 1500 | 1 |
| *Figure 1—figure supplement 3C* | 1 | 100 | 0.1 | $3\times10^5$ | 300 | 1 |
| *Figure 1—figure supplement 3C* | 1 | 100 | 0.02 | $3\times10^5$ | 60 | 1 |

## Constraining biochemical parameters for simulations

Experimental plus-end growth rates were recapitulated using biochemical parameters obtained in a prior study ($k_{on}^{plus} = 0.74$ µM⁻¹ s⁻¹, $K_D^{longitudinal} = 86$ µM, $K_D^{corner} = 25$ nM; *Cleary et al., 2022b* ). Simulations of the minus-end used the same $K_D^{longitudinal}$ and $K_D^{corner}$ as for the plus-end. Iterative fitting in MATLAB was used to optimize an on-rate constant ($k_{on}^{minus}$) that could best recapitulate experimentally observed minus-end growth rates. Each fitting attempt used 50 independent simulations, 300 s in length, of minus-end growth at the same concentrations used for measurements of GMPCPP microtubules.

For all other simulations, 50 independent simulations of 600 s were run for each condition tested. Mixed nucleotide simulations in *Figure 1* were performed at 1 µM total [αβ-tubulin], with varying percentages of GDP-tubulin (up to 20%, in 5% increments). Mixed nucleotide simulations in *Figures 4 and 5* were performed at 1.25 µM total [αβ-tubulin] to mimic experimental conditions, with varied percentages of GDP-tubulin (up to 50%, in 5% increments). Simulations with GDP-tubulin used a 'GDP weakening factor' comparable to one used previously (*Cleary et al., 2022b*), such that the GDP longitudinal interface was 3000- (in *Figure 1*, where we used arbitrary affinities) or 3500-fold weaker (in *Figures 4 and 5*, where we fit affinities to recapitulate growth rates) than the GMPCPP-longitudinal interface; this magnitude weakening is consistent with the large difference between depolymerization rates of GMPCPP and GDP microtubules. Additional simulations in *Figure 1—figure supplements 2 and 3* used a GDP weakening factor such that the GDP longitudinal interface was 300-fold or 30,000-fold weaker than the GTP longitudinal interface.

Simulation parameters and calculated rates for *Figures 1–5* are summarized in *Tables 1 and 2*. Parameters and calculated rates for Supplemental Figures are summarized in *Tables 3 and 4*.

## Acknowledgements

This study was supported by NIH R01-GM135565 to LMR, and by NIH R35-GM139568 to WOH. JMC received support from NIH-T32 GM108563, and LAM received support from a Postdoctoral Research Fellowship in Biology from the NSF (Award 2209298).

## Additional information

### Funding

| Funder | Grant reference number | Author |
|---|---|---|
| National Institute of General Medical Sciences | R01-GM135565 | Luke M Rice |
| National Institute of General Medical Sciences | R35-GM139568 | William O Hancock |
| National Institute of General Medical Sciences | T32-GM108563 | Joseph M Cleary |
| National Science Foundation | PRFB 2209298 | Lauren A McCormick |

The funders had no role in study design, data collection and interpretation, or the decision to submit the work for publication.

### Author contributions

Lauren A McCormick, Conceptualization, Software, Investigation, Visualization, Methodology, Writing - original draft, Writing – review and editing; Joseph M Cleary, Conceptualization, Software, Investigation, Visualization, Writing - original draft, Writing – review and editing; William O Hancock, Luke M Rice, Conceptualization, Supervision, Funding acquisition, Writing – review and editing

### Author ORCIDs

Lauren A McCormick http://orcid.org/0000-0001-9164-0932
Joseph M Cleary http://orcid.org/0000-0003-0879-2543
William O Hancock https://orcid.org/0000-0001-5547-8755
Luke M Rice https://orcid.org/0000-0001-6551-3307

Reviewer #1 (Public Review): https://doi.org/10.7554/eLife.89231.3.sa1
Reviewer #2 (Public Review): https://doi.org/10.7554/eLife.89231.3.sa2
Author Response https://doi.org/10.7554/eLife.89231.3.sa3

## Additional files

### Supplementary files

• MDAR checklist

### Data availability

Source data files have been provided for Figures 1-5.

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
