## [Editor Report · eLife assessment]

This **important** study combines in vitro experiments with simulations to identify the mechanisms governing modulation of microtubule dynamics by GTP hydrolysis. The authors introduce a **convincing** new approach by using a mixed GDP/GMPCPP lattice and varying GDP concentration to reveal that the nucleotide at the interface of two tubulin dimers determines the strength of the interaction between two dimers. Overall, the findings will be of interest to biophysicists and cell biologists, especially in the field of microtubule biology.

---

## [Referee Report · Reviewer #1 (Public Review)]

This study addresses the fundamental question of how the nucleotide, associated with the beta-subunit of the tubulin dimer, dictates the tubulin-tubulin interaction strength in the microtubule polymer. This problem has been a topic of debate in the field for over a decade, and it is essential for understanding microtubule dynamics.

McCormick and colleagues focus their attention on two hypotheses, which they call the "self-acting" model and the "interface-acting" model. Both models have been previously discussed in the literature and they are related to the specific way, in which the GTP hydrolysis in the beta-tubulin subunit exerts an effect on the microtubule lattice. The authors argue that the two considered models can be discriminated based on a quantitative analysis of the sensitivity of the growth rates at the plus- and minus-ends of microtubules to the concentration of GDP-tubulins in mixed nucleotide (GDP/GMPCPP) experiments. By combing computational simulations and in vitro observations, they conclude that the tubulin-tubulin interaction strength is determined by the interfacial nucleotide.

The major strength of the paper is a systematic and thorough consideration of GDP as a modulator of microtubule dynamics, which brings novel insights about the structure of the stabilizing cap on the growing microtubule end.

---

## [Referee Report · Reviewer #2 (Public Review)]

In their manuscript, McCormick, Cleary et al., explore the question of how the nucleotide state of the tubulin heterodimer affects the interaction between adjacent tubulins. They use a solid combination of biochemical reconstitution assays and modeling to reveal that the nucleotide at the interface of two tubulin dimers determines the strength of the interaction between two dimers. Overall, the findings will be valuable to the field of microtubule biology.

---

## [Author Response]

The following is the authors’ response to the original reviews.

We thank the reviewers and editors for their thoughtful assessment and critiques. As detailed below in the point-by-point replies, we have modified the text and figures to clarify points of ambiguity and to document statistical significance in places where we had inadvertently neglected to do so. The manuscript is clearer and more rigorous as a result of the review process.

**Reviewer #1 (Public Review):**
This study addresses the fundamental question of how the nucleotide, associated with the beta-subunit of the tubulin dimer, dictates the tubulin-tubulin interaction strength in the microtubule polymer. This problem has been a topic of debate in the field for over a decade, and it is essential for understanding microtubule dynamics.McCormick and colleagues focus their attention on two hypotheses, which they call the "self-acting" model and the "interface-acting" model. Both models have been previously discussed in the literature and they are related to the specific way, in which the GTP hydrolysis in the beta-tubulin subunit exerts an effect on the microtubule lattice. The authors argue that the two considered models can be discriminated based on a quantitative analysis of the sensitivity of the growth rates at the plus- and minus-ends of microtubules to the concentration of GDP-tubulins in mixed nucleotide (GDP/GMPCPP) experiments. By combing computational simulations and in vitro observations, they conclude that the tubulin-tubulin interaction strength is determined by the interfacial nucleotide.The major strength of the paper is a systematic and thorough consideration of GDP as a modulator of microtubule dynamics, which brings novel insights about the structure of the stabilizing cap on the growing microtubule end.I think that the study is interesting and valuable for the field, but it could be improved by addressing the following critical points and suggestions. They concern (1) the statistical significance of the main experimental finding about the distinct sensitivity of the plus- and minus-ends of microtubules to the GTP-tubulin concentration in solution, and (2) the validity of the formulation of the "self-acting" model with an emphasis solely on the longitudinal bonds.

We thank the reviewer for the comment about statistical significance, and we regret our oversight to have not included that analysis in the original manuscript. We have now included an analysis of statistical significance for the main experimental results supporting the interface-acting model (Fig. 2C and the replotting of those data against a different abscissa in Fig. 3C,D), and more broadly we have ensured that all figure legends contain information about the number of measurements and whether error bars indicate SD or SEM.

The reviewers comment about the sole emphasis on longitudinal bonds helped us realize that a change to Fig. 1 (where we illustrate the two models) would improve clarity. We had originally chosen to illustrate Figure 1 using ‘pure’ longitudinal interactions (with no lateral contacts), and this may be what triggered the reviewer’s comment. We have now revised the figure to show ‘corner’ (longitudinal + lateral) interactions. There are two main reasons for this decision. First, the corner interactions are more long-lived and therefore more important for the phenomena under study. Second, because illustrating corner interactions provides a better basis for us to discuss what is a subtle aspect of our model – that the ‘GDP penalty’ affecting longitudinal or lateral interactions in a corner site is completely equivalent. Thus, our model is not quite as narrow/exclusive as the reviewer suggested. We appreciate having had the chance to clarify this.

**Reviewer #2 (Public Review):**
McCormick, Cleary et al., explore the question of how the nucleotide state of the tubulin heterodimer affects the interaction between adjacent tubulins.(1) The setup of the authors' model, which attributes the dynamic properties of the growing microtubule only to the differences in interface binding affinities, is unrealistic. They excluded the influence of the nucleotide-dependent global conformational changes even in the 'Self-Acting Nucleodide' model (Fig. 1A). As the authors have found earlier, tubulin in its unassembled state may be curved irrespective of the species of the bound nucleotide (Rice et al., 2008, doi: 10.1073/pnas.0801155105), but at the growing end of microtubules, the situation could be different. Considering the recently published papers from other laboratories, it may be more appropriate to include the nucleotide-dependent change in the tubulin conformation in the Self-Acting Nucleotide model.

We understand the reviewer’s perspective, which may be summarized as: “We know conformational changes are happening and that they affect tubulin:tubulin interactions, so why isn’t your model trying to account for that?” In text added to the revised manuscript, we address this critique in the following ways. First, there is not a consensus in the field about how to parameterize the different conformations of tubulin and how they influence tubulin:tubulin interactions. Second, any attempt to explicitly account for different conformations of tubulin would substantially increase the number of adjustable model parameters, which in turn makes the fitting to growth rates more complicated. Third, compared to traditional ‘dynamics’ assays that use GTP, using mixtures of GMPCPP and GDP simplifies the biochemistry by eliminating GTPase. This results in a more uniform composition of nucleotide state in the body of the microtubule polymer, which diminishes the importance of explicitly modeling nucleotide-influenced changes in conformation. Fourth, it seems likely that different conformations of tubulin will modulate both longitudinal interactions (as tubulin becomes straighter the longitudinal contact area grows larger) and lateral interactions (as tubulin becomes straighter, the lateral contact areas on α- and β-tubulin come into better alignment). Our model treats longitudinal and corner (defined as longitudinal + lateral) interactions as independent, so in principle it could be implicitly capturing some of these conformational effects. By refining the strengths of the longitudinal and corner interactions independently, the model effectively allows the strength of longitudinal contacts to be different for pure longitudinal and corner interactions, which might implicitly capture some variations in longitudinal contacts for different tubulin conformations. Our model treats ‘bucket’-type sites (one longitudinal and two lateral interactions) as simply having an additional lateral interaction of equal strength as the first, but because bucket sites have such a high affinity, they rarely dissociate and this small oversimplification is unlikely to have a substantial effect. We have introduced text in several places (bottom of p. 7 and elsewhere) to cover these points.

(2) The result that the minus end is insensitive to GDP (Fig. 2) was previously published in a paper by Tanaka-Takiguchi et al. (doi: 10.1006/jmbi.1998.1877). The exact experimental condition was different from the one used in Fig. 2, but the essential point of the finding is the same. The authors should cite the preceding work, and discuss the similarities and differences, as compared to their own results.

Thank you for reminding us of this paper! We agree that it is an ‘on target’ citation, and have cited and discussed it in the revised manuscript (last paragraph of Introduction, third paragraph of Discussion).

**Reviewer #1 (Recommendations For The Authors):**
1. In my opinion, the way in which the authors have depicted their "self-acting" model in Fig. 1 and in Supplementary Figure 1, makes the model look intuitively implausible. The drawings seem to imply that at the plus-end the GTP hydrolysis in the beta-tubulin subunit somehow allosterically affects the alpha-tubulin subunit of the same dimer to weaken its longitudinal bond with adjacent tubulin dimer. Conversely, at the minus end, the same reaction now affects the very same beta-tubulin subunit, and modulates its longitudinal interaction with the next dimer.However, a more realistic formulation of the "self-acting" model would be that the exchangeable nucleotide affects the lateral bonds, formed by the same beta-tubulin with its lateral neighbors. Although the experimental data in this regard are controversial, at least some supporting evidence for this idea comes from structural arguments, e.g. [Manka, S.W., Moores, C.A. Nat Struct Mol Biol 25, 607-615 (2018).] This "lateral selfacting", but not the "longitudinal self-acting" hypothesis, seems more natural, and it was the one previously implemented in the seminal paper by [Vanburen et al, 2002 Proceedings of the National Academy of Sciences 99.9 (2002): 6035-6040.] and later by other some other models as well.

This point has been addressed above, in part by modifying the cartoon in Fig. 1.

2. To better clarify, which exact models are considered in this manuscript, it would be helpful if the authors provided a detailed table with all simulation parameters, including, k_off_loner, k_off_bucket and k_off_corner, for both nucleotide states, in both the selfacting and the interface-acting models.

Thank you for the suggestion. We have added tables that show all simulation parameters, as well as the corresponding calculated on- and off-rates for each interaction.

3. I am not sure that using some 'arbitrarily chosen' parameters is very helpful in Chapter 1 of Results. In fact, the results, obtained with an unconstrained set of parameters may be misleading or provide ambiguous answers. In other words, how reliable are the conclusions, based on the arbitrary parameter set? For example, could the dependences of the microtubule growth rate on the GDP-tubulin content be more or less pronounced with a different set of arbitrarily chosen parameters, compared to the graphs in Fig. 1BC?

This is a fair criticism. In response, we have added three new sets of simulations that each test different choices of the biochemical parameters used in Figure 1. With respect to the original parameters, we tested a weaker and stronger choice for the longitudinal interaction (KDlong, a 100-fold range), the corner interaction (KDcorner, a 25-fold range), and the GDP weakening factor (a 100-fold range). The predicted supersensitivity of plus-end growth rates to GDP in the self-acting vs interface-acting mechanisms is robust across the range of different choices for the above parameters (Figure 1 Supplements 1 and 2). Parameters for these new simulations are shown in Tables 3 and 4.

4. It took me some time to comprehend why the minus-end growth rate is assumed to be dependent only on the concentration of the GMPCPP-tubulin (in section 2 of Results). It would be great if the authors simply plotted the simulated dependence of the growth rate on the GMPCPP-tubulin concentration in the case when no GDP-tubulin was added. As I understand, that curve should almost exactly match the dependence observed in Fig 1B, correct? Otherwise, it does not seem obvious, why GDP-tubulin does not impede the minus-end growth. Again, is this conclusion model- and parameterdependent? This question is related to point 3 above.

The minus-end growth rates decrease in proportion to the concentration of GMPCPPtubulin. We have added a note on minus-end growth rates in the Figure 1 legend.

5. I was not quite convinced by the evidence for distinct sensitivities of the plus- and minus-end growth rates to GDP-tubulin concentration (Figure 2C and Fig 3C, D). These are the key experimental measurements in the paper. Therefore, I suggest that the authors try to strengthen this point by additional measurements to increase statistics. Or at least, please, explain the data points, the error bars, and provide some information on the number of independent measurements and the statistical significance between the curves. Maybe, they could be directly compared after normalizing by the "all GMPCPP growth rate"? How was the "1.5-fold" ratio obtained in Fig 2C? Does that number refer only to a certain GDP-tubulin concentration or does that value somehow characterize the whole range of the concentrations measured?

This has been addressed above.

**Reviewer #2 (Recommendations For The Authors):**
(1) The setup of the authors' model, which attributes the dynamic properties of the growing microtubule only to the differences in interface binding affinities, is unrealistic. They excluded the influence of the nucleotide-dependent global conformational changes even in the 'Self-Acting Nucleodide' model (Fig. 1A). As the authors have found earlier, tubulin in its unassembled state may be curved irrespective of the species of the bound nucleotide (Rice et al., 2008, doi: 10.1073/pnas.0801155105), but at the growing end of microtubules, the situation could be different. Considering the recently published papers from other laboratories, it may be more appropriate to include the nucleotide-dependent change in the tubulin conformation in the Self-Acting Nucleotide model.(2) The result that the minus end is insensitive to GDP (Fig. 2) was previously published in a paper by Tanaka-Takiguchi et al. (doi: 10.1006/jmbi.1998.1877). The exact experimental condition was different from the one used in Fig. 2, but the essential point of the finding is the same. The authors should cite the preceding work, and discuss the similarities and differences, as compared to their own results.

These look identical to above and were addressed there.